# Carbon fixation and rhodopsin systems in microbial mats from hypersaline lakes Brava and Tebenquiche, Salar de Atacama, Chile

Daniel Kurth[1], Dario Elias[2¤], María Cecilia Rasuk[1], Manuel Contreras[3], María Eugenia Farías[1]*

**1** Laboratorio de Investigaciones Microbiológicas de Lagunas Andinas (LIMLA), Planta Piloto de Procesos Industriales Microbiológicos (PROIMI), CCT, CONICET, Tucumán, Argentina, **2** Facultad de Ingeniería, Universidad Nacional de Entre Ríos, Oro Verde, Entre Ríos, Argentina, **3** Centro de Ecología Aplicada (CEA), La Reina, Santiago, Chile

¤ Current address: ECLAMC (Estudio Colaborativo Latinoamericano de Malformaciones Congénitas) at CEMIC (Centro de Educación Médica e Investigaciones Clínicas), CONICET, Buenos Aires, Argentina
* mefarias2009@gmail.com

**Data Availability Statement:** Raw sequence data have been deposited in the ENA European Nucleotide Archive (ENA) under the accession number ERP107533. Filtered reads for several

## Abstract

In this work, molecular diversity of two hypersaline microbial mats was compared by Whole Genome Shotgun (WGS) sequencing of environmental DNA from the mats. Brava and Tebenquiche are lakes in the Salar de Atacama, Chile, where microbial communities are growing in extreme conditions, including high salinity, high solar irradiance, and high levels of toxic metals and metaloids. Evaporation creates hypersaline conditions in these lakes and mineral precipitation is a characteristic geomicrobiological feature of these benthic ecosystems. The mat from Brava was more rich and diverse, with a higher number of different taxa and with species more evenly distributed. At the phylum level, Proteobacteria, Cyanobacteria, Chloroflexi, Bacteroidetes and Firmicutes were the most abundant, including ~75% of total sequences. At the genus level, the most abundant sequences were affilitated to anoxygenic phototropic and cyanobacterial genera. In Tebenquiche mats, Proteobacteria and Bacteroidetes covered ~70% of the sequences, and 13% of the sequences were affiliated to *Salinibacter* genus, thus addressing the lower diversity. Regardless of the differences at the taxonomic level, functionally the two mats were similar. Thus, similar roles could be fulfilled by different organisms. Carbon fixation through the Wood-Ljungdahl pathway was well represented in these datasets, and also in other mats from Andean lakes. In spite of presenting less taxonomic diversity, Tebenquiche mats showed increased abundance and variety of rhodopsin genes. Comparison with other metagenomes allowed identifying xantorhodopsins as hallmark genes not only from Brava and Tebenquiche mats, but also for other mats developing at high altitudes in similar environmental conditions.

Andean metagenomes are available in ENA Study PRJEB19379. Assembled contigs are available in MG-RAST with IDs mgm4576367.3 (Brava) and mgm4576401.3 (Tebenquiche)

**Funding:** Financial support was provided to MEF and MC, by projects PICT V Bicentenario 2010-1788, PICT V 2015-3825 from Fondo para la Investigación Científica y Tecnológica (FONCYT), Argentina (https://www.argentina.gob.ar/ciencia/agencia/fondo-para-la-investigacion-cientifica-y-tecnologica-foncyt), Sociedad Química y Minera de Chile (https://www.sqm.com/), and Centro de Ecología Aplicada (https://cea.cl/). DK, MCR and MEF are CONICET fellows. The funders had no role in study design, data collection and analysis, decision to publish, or preparation of the manuscript.

**Competing interests:** The authors have declared that no competing interests exist.

## Introduction

The Atacama desert, located in northern Chile, has been termed "the driest location in the world" and it's well known from many studies investigating the dry limits of life, and as a model for the search of life in Mars [1]. Those studies refer mostly to the hyperarid core of the desert, but the region comprises diverse environments, including high plateaus, *bofedales* (peatlands), geyser fields, water streams, hypersaline lakes, and salt flats.

Particularly, the Salar de Atacama is one the largest evaporitic basins in Chile, encompassing an area of around 3000 km$^2$ [2], with an altitude of around 2500 m. Within this large area, a number of lakes are found, including: Laguna de Piedra, Laguna Tebenquiche, Chaxas, Burro Muerto and La Brava [3]. Extreme conditions define this region, including intense solar radiation, wide diel temperature variations, hypersalinity due to net evaporation, and high arsenic concentrations. This combination of extreme conditions is found throughout the whole Andean region known as Puna, comprising a high altitude plateau in Central Andes with a mean altitude of 3700 m [4]. These conditions and the presence of a water interface are prone to microbial mat development [5, 6], and the lakes found in this region, collectively referred as High Altitude Andean Lakes (HAAL) usually present diverse microbial mats [7, 8]. Early studies of mats in HAAL featured photosynthetic organisms as they were considered fundamental to the systems [9–11]. In recent years, the application of next generation sequencing has allowed to taxonomically characterize many types of HAAL mats, revealing an unexpected diversity [7, 12–16]. WGS analysis provided further insight in these systems, highlighting the importance of ancient metabolisms and their relevance as Early Earth models [17–19].

Besides the extreme conditions, light is a key element for the occurrence of microbial mats. Mats developing in hypersaline conditions, such as salterns or the HAAL, are phototrophic [20]. They have a layered structure, where microorganisms populate different niches, according to their preferences, generating and maintaining physicochemical gradients [21]. Cyanobacteria lay the foundation of the community, with their role as primary producers [22], but they are not the only phototrophic members. The photic zone of the mats, also includes anoxygenic photosynthesizers and photoheterotrophs. Phototrophic microorganisms transduce light into energy by two general systems: chlorophyll related pigments and retinal-based microbial rhodopsins [23]. Primary production in microbial mats is due to photosynthesis, and the role of microbial rhodopsins is not clear.

The rhodopsin family includes light sensors, light-gated ion channels, and light-driven ion pumps, with proton pumps related to energy conversion [24]. Proton pumps belong to different subfamilies: proteorhodopsins and xantorhodopsins. It has been suggested that proteorhodopsins play a key role in carbon cycles and energy flux in aquatic microbial systems [23]. In marine microbial ecosystems, where environmental conditions are variable and subjected to periods of starvation, organisms that express proteorhodopsin could have an adaptive advantage. In this scenario, the photoproteins would serve to harvest extra energy by phototrophy [25]. Xantorhodopsins were discovered more recently [26, 27], and could serve similar functions. In addition to retinal, they bear 4-keto-carotenoids antenna pigments. Even though they are not abundant in marine systems, they seem to be more specific of icy environments [28].

In this work, two microbial mats from lakes Tebenquiche and Brava, in Salar de Atacama, were analysed. These lakes show a variety of mat systems, in various lithification states [7, 12, 13]. WGS analysis allowed functional characterization of two non-lithifying mats, and metagenomic comparisons provided insights in the diversity and role of rhodopsins in these mats. Arsenic is a major subject for HAAL systems, and analysis of its metabolism in Brava and Tebenquiche mats has been performed elsewhere [29] and will not be discussed in this work.

## Materials and methods

### Sampling

Samples from non-lithifying mats were obtained from Tebenquiche and La Brava lakes in November 2012. At the moment of sampling the locations were freely accessible, with no protected areas from Chilean government, and no specific permissions were required for these activities. Geographical locations of the sampling points were S 23˚43'48.2", W 068˚14'48.7" for La Brava mat and 23˚08'18.5", W 068˚14'49.9" for Tebenquiche, as shown in Fig 1. In Tebenquiche, samples were taken from the southeastern shoreline water body. Mats had semispheroidal morphologies covered by a pink leathery biofilm (Fig 1A). At the time of sampling, the water column above the mat was 10–20 cm high, with a conductivity of 161 mS.cm$^{-1}$, a temperature of 23.3˚C, and pH 7.8. La Brava sampling point was located on the east of a water input channel (Fig 1B). The surface of the sample consisted of a continuous pink layer covered by 3–5 cm of water during the dry season and 5–10 cm during the wet season. At the time of sampling, the water column had a conductivity of 178 mS.cm$^{-1}$, a temperature of 30.1˚C, and pH 7.8. Triplicate cores 2 cm$^2$ each were taken to a depth of 3 cm of the mat and stored at 4˚C during transportation.

### DNA extraction and sequencing

Metagenomic DNA was extracted from each sample core with the Power Biofilm DNA Isolation Kit (MO BIO Laboratories, Inc.), and pooled prior to sequencing. One round of extraction and pooling allowed obtaining enough DNA (pooled samples included DNA from three extractions, one from each triplicate core). Sequencing was performed at INDEAR genome sequencing facility (Argentina). TruSeq libraries were prepared according to TruSeq ℞ DNA Sample Preparation Guide Illumina (July 2012), starting from 2 μg DNA. Paired end libraries were sequenced with an Illumina HiSeq 1500 instrument. Raw sequence data have been deposited in the ENA European Nucleotide Archive (ENA) under the accession number ERP107533.

### Sequence processing and analysis

Sequencing results were visualized with fastqc (http://www.bioinformatics.babraham.ac.uk/projects/fastqc/). Reads were trimmed and filtered with a custom script, where adapter sequences were identified and removed. Reads with unassigned bases (N) or shorter than 50 bp were discarded. Paired-end reads were merged and converted to fasta format, and assembled with IDBA-UD [30]. Assembly was performed with parameters—mink 35—maxk 151—min_pairs 2. Coverage for each contig was obtained by mapping the reads using bowtie2 [31]. Assembled contigs were annotated using the metagenomic analysis server MG-RAST [32], and results are publicly available there, with IDs mgm4576367.3 (Brava) and mgm4576401.3 (Tebenquiche) Raw counts for each gene were calculated with htseq-count, a module of HTSeq [33], normalized with the TPM method [34], and reported as counts per million (CPM).

### Taxonomic analyses

Taxonomic affiliation was obtained from MG-RAST server [32], based on BLAT [35] results against M5nr database [36] for protein sequences obtained from the assembled contigs, with minimum alignment 15, e-value $< 10^{-5}$, and identity $> 60\%$. In addition, reads were annotated by kaiju 1.5.0 [37] against ProGenomes database [38], using Greedy run mode.

An alpha diversity estimation was also provided by the MG-RAST server, based on the taxonomic annotations for the predicted proteins, and computed as the antilog of the Shannon

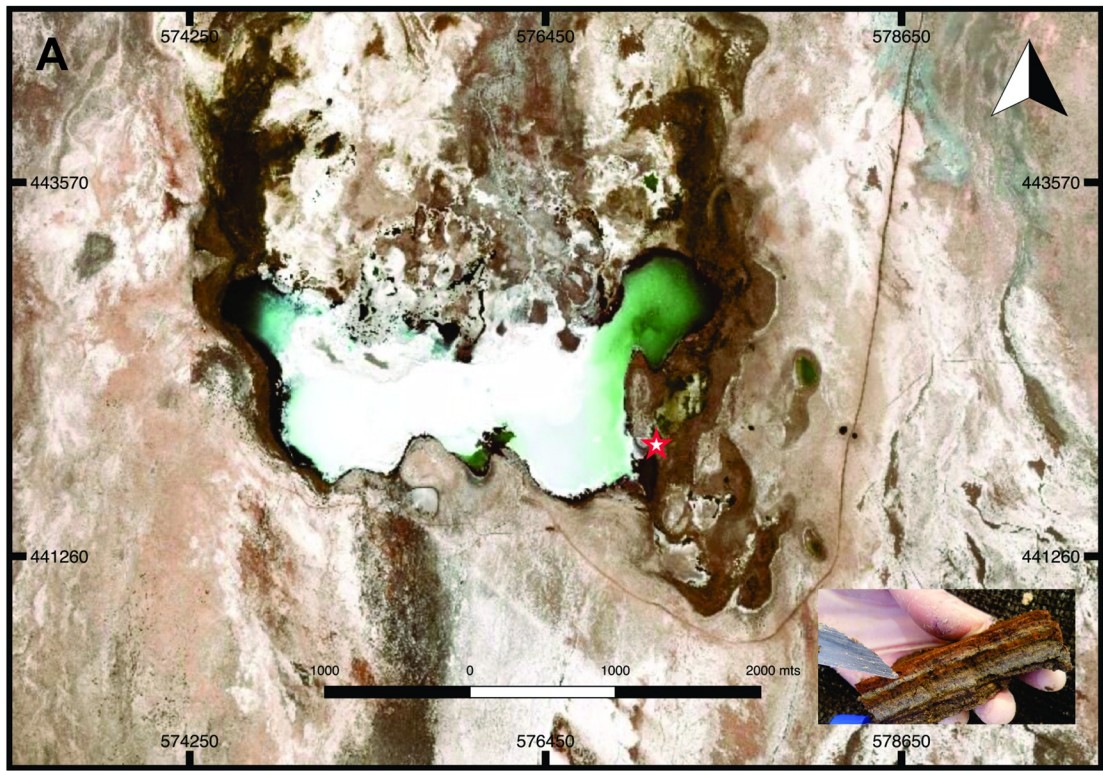

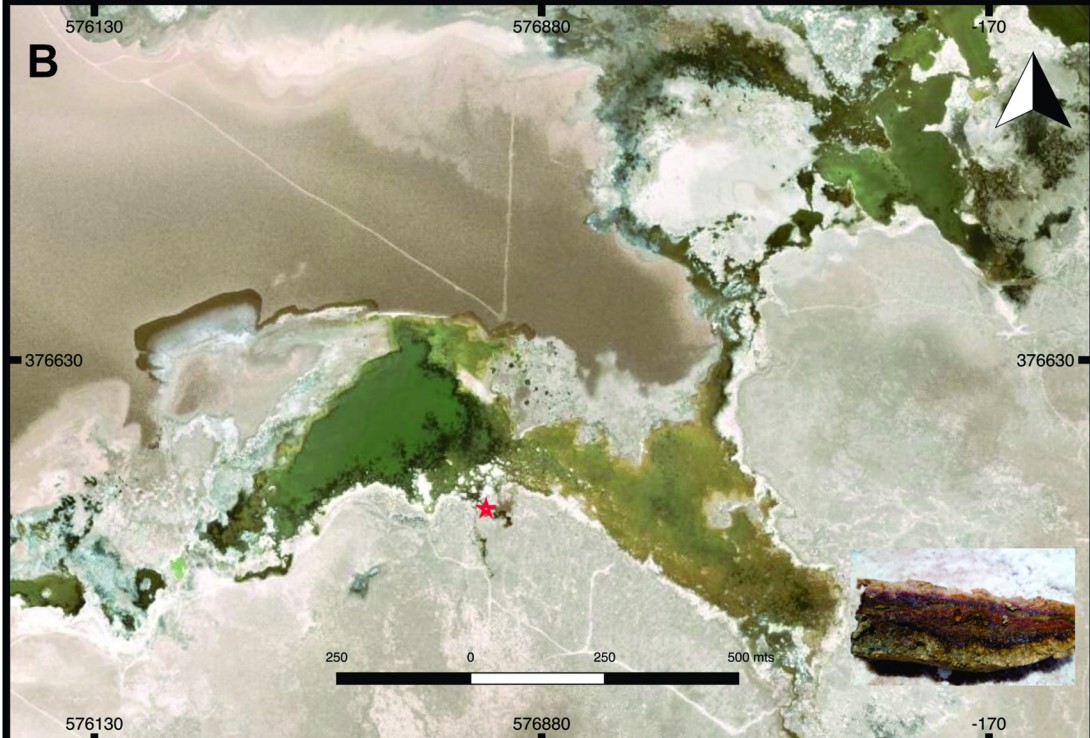

**Fig 1.** Maps of (A) Tebenquiche and (B) Brava Lakes, showing sample collection sites. Inset: Vertical section of the microbial mats. Map data available from the U.S. Geological Survey.

diversity:

$$\text{Alpha diversity estimate} = 10^{-\sum_i p_i \log(p_i)}$$

where pi are the proportions of annotations in each of the species categories. Each p is a ratio of the number of annotations for each species to the total number of annotations.

## Functional genes identification

COG annotation was obtained from MG-RAST API [39]. Selected markers were analyzed, including COG1850 for RuBisCO (Calvin-Benson cycle), COG2301 for citrate lyase (rTCA), COG2368 for 4-hydroxybutyryl-CoA dehydratase (DH/HB-3HP/HB cycles), and COG1152 for acetyl-CoA synthase (Wood-Ljungdahl pathway). KEGG GhostKOALA webservice version 2.0 [40] assigned protein sequences from each assembly to KO identifiers. Processes were defined based on groups of identifiers: anoxygenic photosynthesis (bacteriochlorophyll and *puf* genes: K08928, K08929, K08940, K08941, K08942, K08943, K08944), photosynthesis (*psa* and *psb* genes: K02703, K02704, K02705, K02706, K02707, K02708), sulfate reduction (*aprAB* and *dsrAB* genes: K00394, K00395, K11180, K11181), sulfur oxidation (*sox* genes: K17222, K17223, K17224, K17225, K17226, K17227), nitrogen fixation (*nif* genes: K02588, K02586, K02591, K00531), denitrification (reductases for nitrate, nitrite, nitric oxide and nitrous oxide: K00370, K00371, K00374, K02567, K02568, K00368, K15864, K04561, K02305, K15877, K00376), and ammonia oxidation (*amo* genes: K10944, K10945, K10946, K10535). Sulfate reduction proteins were further checked for the presence of *dsr* sulfur oxidation genes. For this, the DNA sequences were obtained from the contigs, aligned to the *dsr* database [41] and placed in the reference tree usign arb [42]. Carbon fixation pathways marker genes were also retrieved from GhostKOALA annotation (K01601, K01648, K14534, K14138/K00193). All hits were confirmed by NCBI BLAST 2.7.1+ analysis against RefSeq database.

Rhodospsins were identified with HMMer 3.1 (http://hmmer.org/), based on the HMM model from Pfam PF1036, and using 1E-05 as e-value threshold. Taxonomy was reassigned using NCBI BLAST 2.7.1+ against RefSeq database, and parsing results with the lowest common ancestor (LCA) algorithm implementation from MEGAN [43]. Rhodopsin type was determined by ncbi-blast-2.7.1+ against MicRhoDE [44] using the best hit.

## Metagenomic comparisons

MG-RAST API [39] was used to obtain taxonomic RefSeq annotations at phylum and genus levels, and COG level 4 functional annotations. The analysis were performed in R [45]. Data was normalized using DESeq2 [46], and dissimilarity matrices based on Bray-Curtis distances were constructed using vegan [47]. Principal Coordinates analysis was used to visualize sample groupings, and these results were further validated with pvclust [48].

## Rhodopsin/recA/rad51 detection on metagenomic reads

Metagenomes for comparison were retrieved from the early EBI Metagenomics webserver, now MGnify [49]. Biomes were detailed in the downloaded associated metadata, however, for this analysis metagenomes were arbitrarily grouped in the following categories: oceanic, coastal ocean, estuary, hydrothermal vent, lake, soil, salt pan, glacier, Organic Lake (Antarctica), Yellowstone, Puna, and Microbialites. These latter, more specific categories, were included to highlight locations of interest to compare with the mats in study. The server pipeline did not accept raw reads from Brava and Tebenquiche, and preprocessing, which included adaptor trimming and quality filtering, was performed with Trimmomatic [50]. The processed

reads for these metagenomes and also for Socompa [18] are available in ENA Study PRJEB19379.

Metagenomic reads obtained from the webserver were translated to amino acid sequences in the six reading frames and filtered with HMMER 3.1 (http://hmmer.org/) using a MicR-hoDE (MR) profile [44]. Reads with a score lower than 8 were discarded. Filtered reads were assembled with MEGAHIT [51], running the k-mer values from 21 to 141 with a step of 2. Using BBMap (www.sourceforge.net/projects/bbmap/) and SAMtools [52] the contigs that did not align with at least one read with an identity percentage greater than 75% were discarded. Reads not aligned to contigs were included in the following steps as unit contigs. Finally, Frag-GeneScan [53] was used with default parameters to predict gene fragments in contigs.

To classify the fragments and assign them to the rhodopsin family, a method was implemented that evaluates the length, score, coverage and region of the MR profile where each fragment is aligned. Based on a dataset from known rhodopsin fragments, the classifier was defined in the following way: X being the fragment to be classified, this will be considered rhodopsin if its score and coverage are greater than the 0.04 quantile of the score and coverage of the set of positive fragments, which have the length of X and the extremes of their alignment contain those of X, or are contained by those of X (S1 Fig). More details are provided as S1 File. An analog procedure was followed to classify RecA and Rad51 reads.

## Metagenomes clustering

The rhodopsin fragments were located in a tree generated with MR sequences. To do this, a tree was created with FastTree [54] from the multiple alignment of MR, using the MR tree as the initial topology. For the creation of the tree, the JTT [55] and WAG [56] evolutionary models were tested. Through the Robinson-Foulds test implemented in ETE [57], it was determined that the tree most similar to MR was the one generated with JTT. The HMMER hmmalign function was then used to align the fragments to the multiple MR alignment. Finally, the fragments were placed in the tree using PPLACER [58] with the posterior probability method. Only the most probable location of each fragment was considered and those that could not be located in the tree were discarded. In turn, metagenomes that had less than 10 fragments located in the tree were completely excluded. Then the 44 clades closest to the root (RC) of the rhodopsin tree were selected and the proportion of readings by RC was calculated as:

$$p_i = \frac{\sum^{F_i} h/l}{\sum^{F} h/l}$$

where $p_i$ is the proportion in the ith RC, F is the total number of fragments in the tree, $F_i$ is the number of fragments in the ith RC, h is the number of reads related to a fragment, and l is the fragment length.

The metagenomes were grouped according to the proportion by RC, for this a recursive procedure was used that performs a hierarchical grouping from non-hierarchical sub-groups. The procedure was implemented in R [45]. On each iteration RCs were eliminated by three criteria: (i) more than 95% of the metagenomes had a proportion less than 0.001 and the remaining 5% less than 0.02; (ii) metagenomes had the same proportion; (iii) in the case of fully correlated RC, only one remained. Then the best combination of distance functions, aggregation methods and the number of groups (k) is determined. The distance functions were evaluated: Euclidean, Maximum, Manhattan, Canberra and Minkowski (implemented in *dist* R function); aggregation methods: *Ward D*, *Ward D2*, *Simple*, *Complete*, *Average*, *Mcquitty*, *Median*, *Centroid and K-Means* (implemented in *hclust* and *kmeans* R functions);

and the range of k from 2 to 10. Through the NBClust library [59] we used 27 indexes (parameter *all*) to determine the k value and 12 indexes (*kl, ch, ccc, cindex, db, silhouette, ratkowsky, ptbiserial, mcclain, dunn, sdindex, sdbw*) for the combination of distance functions and aggregation methods. The choice was made by a majority of the evaluated indices.

### RHO family's proportion

The proportion of each RHO family by metagenome was determined from the annotations of MR sequences, multiplying the percentage of each family in each RC by the proportion of reads of each metagenome in said RC.

### RHO abundance

RHO counts were normalized against the recombinases (REC) counts. This is based on the assumption that *recA* and *rad51* represent a single copy gene in bacterial and archaeal cells [60]. The REC fragments were identified analogously to the RHO, using the multiple alignments of the RecA and Rad51 Pfam families [61]. In addition, fragment length was also considered for normalization [62].

Rhodopsin abundance was calculated as:

$$\text{RHO abundance} = \sum \text{RLr} / \sum \text{RLe}$$

where $RL_r$ is the ratio of the number of rhodopsin reads to fragment length and $RL_e$ is the ratio of the number of recombinase reads to fragment length.

## Results

### Assembly results

DNA from the mats was obtained and sequenced by shotgun strategy with Illumina technology. Quality filtering yielded over 30 Gbp. IDBA_UD [30] assembled over 1 200 000 contigs for each mat, with N50s 1566 bp and 1394 bp, and total lengths 1414 Mbp and 1264 Mbp for Brava and Tebenquiche, respectively (S1 Table). Reads aligned on each assembly represented 88.87% and 90.08% of the quality filtered reads. Assemblies were uploaded to the MG-RAST server [32] for annotation and further analysis. Gene annotation on these contigs predicted 1 723 512 proteins for Brava mat (BM) and 1 574 837 for Tebenquiche mat (TM).

### Taxonomic and functional diversity

Taxonomy was assigned from MG-RAST RefSeq annotations of the assemblies and from raw reads assignments with Kaiju [37]. Kaiju assigned 45% and 51% of the reads from Brava and Tebenquiche, respectively, while all contigs were assigned by MG-RAST, even though with both methods these assignments included a small number of "Unclassified Bacteria" (up to 3.6% for kaiju in Tebenquiche). Nevertheless, the profiles obtained were similar between methods, as shown on Fig 2. Both mats were dominated by Proteobacteria and Bacteroidetes, with other major phyla including Chloroflexi, Cyanobacteria, Firmicutes, Actinobacteria, and Spirochaetes. Chloroflexi, Cyanobacteria, and Firmicutes were much more abundant in BM. At the genus level, in TM the most abundant included the Bacteroidetes *Salinibacter* (13.2%), *Rhodothermus* (3.2%), and *Bacteroides* (1.7%), and the Alphaproteobacteria *Rhodospirillum* (1.9%). In BM, the Chloroflexi *Roseiflexus* (6.5%) and *Chloroflexus* (3.5%) were the most abundant, followed by the Cyanobacteria *Microcoleus* (3.5%), *Cyanothece* (2.7%), *Nostoc* (2.1%), and *Oscillochloris* (1.9%).

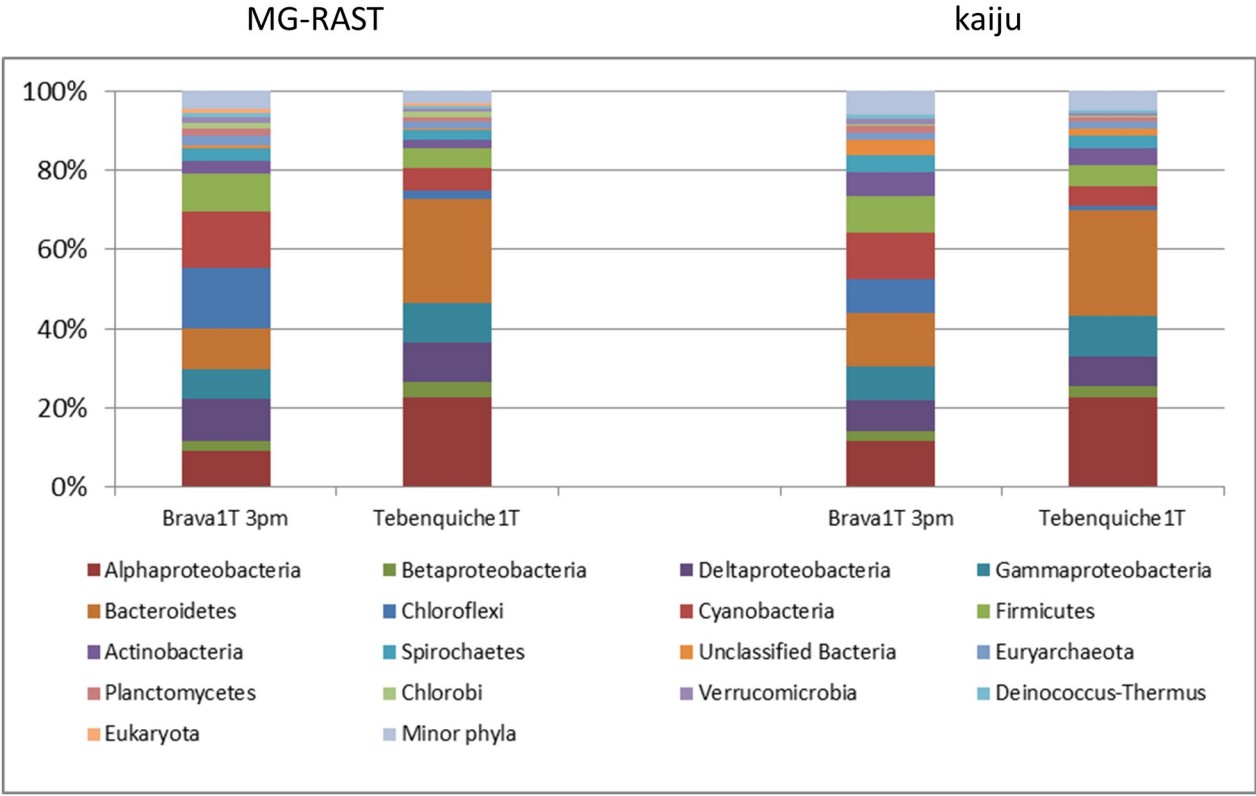

**Fig 2. Taxonomic abundance at phylum level on each mat.** Annotations based on MG-RAST or kaiju (see Materials and Methods for details). Proteobacteria were annotated at class level.

Diversity was higher in Brava, with an α-diversity value of 572 observed species compared to 410 for Tebenquiche.

A glimpse on the biogeochemistry of these mats was obtained through analysis of genes involved in several processes, including photosynthesis, carbon, nitrogen and sulfur cycles (Fig 3A). Similar functional roles could be observed in the mats, with a complete sulfur cycle represented on each, and nitrogen cycles lacking ammonia oxidation. Relative abundances were different, with oxygenic photosynthesis dominating in BM, and all other processes more abundant in TM (S2 Table).

Sulfate is an abundant ion in these hypersaline lakes [7, 12, 13, 63]. Sulfur cycling might have a role in lithification for these systems. Genes representing sulfur oxidation and reduction were identified in both datasets. *apr* and *dsr* genes for sulfate and sulfite reduction, respectively, were both present at similar abundances. They affiliated mostly to Deltaproteobacteria, being most abundant the family Desulfobacteraceae, and a small proportion was affiliated to Firmicutes. From our analysis we could not confirm that both genes are present on the same organisms, but taxonomic affiliations up to the family level are similar. Some *dsr* genes are able to catalyze oxidation and reduction [41], however, comparison to the DsrAB database identified only 5 genes affiliated to oxidative processes in TM and none in BM. Sulfur oxidation was mainly performed by Alphaproteobacteria, including Rhizobiales and Rhodobacterales. In BM sulfur oxidizer Euryarcheota from the methanogenic Methanomicrobiales order are also present, and could be related to the presence of methylotrophic *Methylobacterium*.

Nitrogen cycle genes representing nitrogen fixation, nitrification and denitrification were identified, including nitrogenase, reductases for nitrate, nitrite, nitric oxide and nitrous oxide

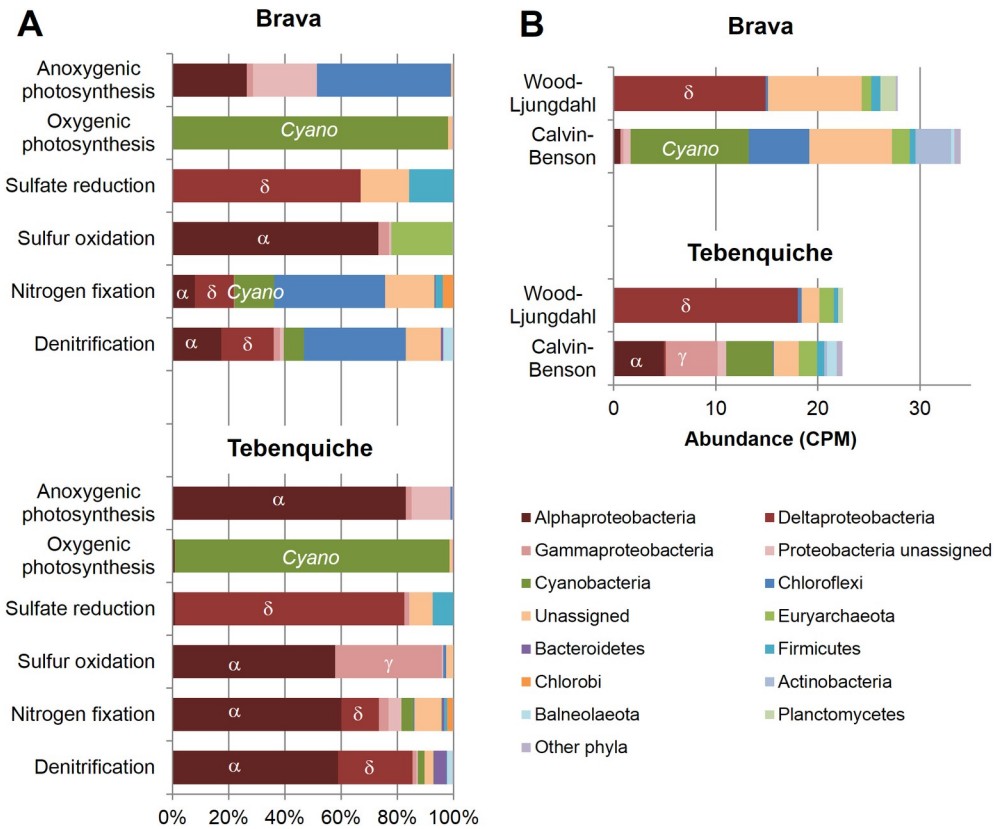

**Fig 3. Functional analysis of biogeochemical cycles, based on KO markers.** (A) Relative abundance and taxonomic affiliation of markers associated to each process. (B) Abundance (in counts per million–CPM) and taxonomic affiliation of markers for different carbon fixation pathways. Additional markers analyzed had very low abundances or provided unreliable results (see text).

and ammonia oxidation (*amo*) genes. Ammonia oxidation was absent in BM and very scarce in TM. Nitrogen fixation and denitrification genes were abundant in both mats, but more important in TM. Chloroflexi had a major functional role in nitrification and denitrification in BM, with Cyanobacteria, Deltaproteobacteria and Alphaprotebacteria, and other minor phyla involved. However, in TM these roles were dominated by Alphaproteobacteria from Rhodobacteraceae family.

Oxygenic photosynthesis was almost exclusively carried out by Cyanobacteria, and more than 85% of the sequences assigned to this phyla could not be affiliated to higher taxonomic levels. In BM, half of the genes assigned to anoxygenic photosynthesis were affiliated to Chloroflexi, mostly *Choroflexus* (81.5%), and diverse Alphaproteobacteria including Rhizobiales and Rhodobacterales. Instead, in TM anoxygenic photosynthesis was carried out almost exclusively by Alphaproteobacteria, mostly Rhizobiales but also Sphingomonadales, Rhodobacterales and Rhodospirillales. Interestingly, 27% of the genes associated to anoxygenic photosynthesis were affiliated to *Methylobacterium* genus. These organisms are known pigmented methylotrophs and bear photosystem genes, but have been only seldom related to anoxygenic photosynthesis [64, 65].

Abundance of markers from carbon fixation highlights the importance of alternative pathways (Fig 3B). It is striking that chemoautotrophic Wood-Ljungdahl pathway bear similar number of hits than Calvin-Benson cycle markers. Other pathway markers, such as ATP-

citrate lyase markers (K15230, K15231) for rTCA cycle, showed normalized abundances below 1 CPM. From Calvin-Benson pathway, taxonomic affiliation indicates that Cyanobacterial markers are only 20% in TM and 34% in BM. Other well-known anoxygenic photosynthesizers also fix carbon through Calvin-Benson cycle, such as Chloroflexi in BM (17%) and members of Alphaproteobacteria (22%) in TM. Thus, around 50% of the Calvin-Benson markers present in both mats belong to less known potential carbon fixers, for example Actinobacteria (10%). In addition, a large number of Wood-Ljungdahl pathway markers are affiliated to Deltaproteobacteria, 50% in BM and 80% in TM. This might be indicative of alternative primary production by either of these pathways, although carbon fluxes cannot be determined by this methodology. Finally, 4-hydroxybutyryl-CoA dehydratase (K14534), the key enzyme for both the 3-hydroxypropionate/4-hydroxybutyrate (HP/HB) and the dicarboxylate/4-hydroxybutyrate (DC/HB) cycles, was apparently well represented in the dataset. However, many hits pointed to related enzymes from the same family, unrelated to carbon fixation (S3 Table).

A few model mat systems were used for comparison at both taxonomic and functional levels. These included hypersaline mats from Shark Bay (Australia) [66], marine mats from Highbourne Cay (Bahamas) [67] stromatolites from Cape Recife and Schoenmakerskop (South Africa) [68], oligotrophic mats from Cuatro Ciénagas (Mexico) [69], and two HAAL mats: a rare archaeal biofilm from Diamante lake [19] and the hypersaline mats from Socompa [18]. Functional analysis based on COG markers for carbon fixation confirmed the KEGG-based observations for BM and TM. In this analysis, rTCA cycle was not included, as COG2301 is not specific for ATP-citrate lyase, recognizing also the enzyme from TCA cycle and overestimating rTCA abundance. The presence of Wood-Ljungdahl pathway only in the HAAL mats (Brava, Tebenquiche, Socompa) is quite remarkable (red columns in Fig 4). It is also present in Shark Bay mats, as previously reported [70]. Principal components analysis based on taxonomic data at phylum and genus level and COG functional data supports clustering of Shark Bay mats with BM, TM and Socompa HAAL mats (S2 Fig).

## Comparative analysis of rhodopsins

In oligotrophic environments such as the HAAL, alternative energy-generating systems could be important for survival. This could favor the presence of rhodopsins to harvest extra energy [25, 71].

In Brava and Tebenquiche microbial mats, 182 and 253 coding sequences were found for rhodopsins in the metagenomic assemblies, respectively, pointing to a broader diversity in Tebenquiche. Not only that, but abundances were also higher in Tebenquiche. The sequences were classified according to Micrhode database [44], with the most represented type being xanthorhodopsin, followed by NQ- and halorhodopsins (Fig 5). Sensory rhodopsins were represented by only a few sequences. The taxonomic distribution was biased towards a few phyla, Bacteroidetes accounted for over half of the observed rhodopsins, and sequences from Cyanobacteria, Alphaproteobacteria and Betaproteobacteria were relatively abundant.

## Rhodopsin metagenomic comparisons

Given that rhodopsins are relatively abundant in these mats, it was hypothesized that they could be a feature related to the extreme conditions. Besides Brava and Tebenquiche, microbial rhodopsins had already been observed in HAAL isolates [71–73]. To test this idea, 1063 metagenomes (Identifiers are provided in S4 Table) from a wide range of environments were compared, including quality filtered datasets from HAAL, deposited in ENA: Brava (ERR1878617), Tebenquiche (ERR1878606), Diamante [19] (ERR1824222), Socompa [18] (ERR1924357) and

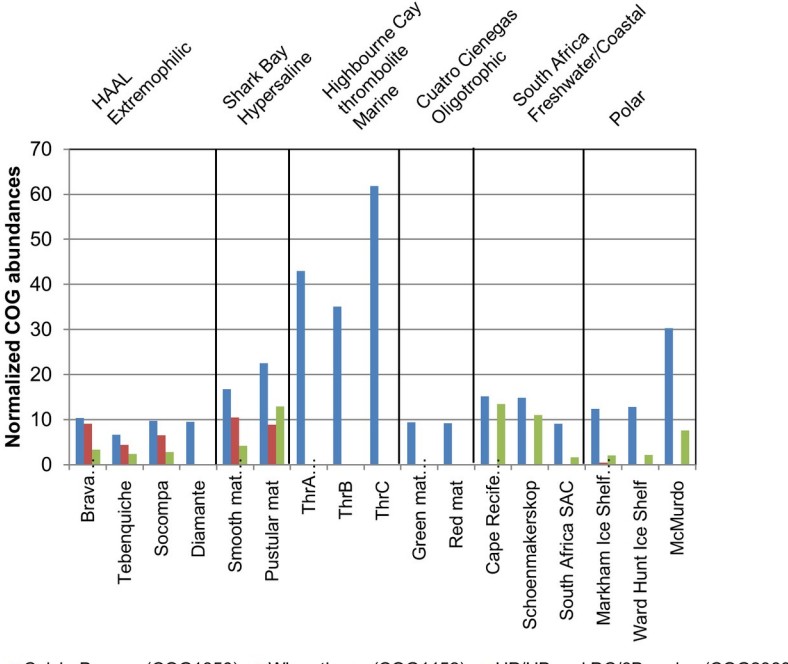

**Fig 4. COG markers for carbon fixation in different metagenomes.** Abundance and taxonomic affiliation of COG markers for different carbon fixation pathways were obtained from MG-RAST. Data for Brava and Tebenquiche is compared to several microbial mats and microbialites.

Llamara (ERR1937966). The comparison was based on rhodopsin fragment identification and assembly in raw reads obtained from EBI metagenomes [49].

Rhodopsin abundances were high in aquatic environments and low in soil environments. HAAL environments ranked in between, but abundances were variable (Fig 6). Metagenomes were grouped based on phylum abundance (S3 Fig) and rhodopsin family. For these comparisons, only 673 metagenomes with more than 10 rhodopsin fragments were used. Interestingly, only when grouped by rhodopsin type, Brava and Tebenquiche metagenomes grouped together, but the group also included metagenomes from other HAAL (Socompa and Llamara), glacier, and the antarctic Organic lake. Even though the metagenomes group together based on the abundance of several rhodopsin clades, their main characteristic was the increased abundance of xantorhodopsin (Fig 7A). This could define a hallmark of HAAL locations, and more broadly, relate xantorhodopsins to sites with extreme, oligotrophic conditions.

Xantorhodopsins are also abundant in group 2 and group 3 metagenomes (Fig 7B), which include coastal area sediments, antartic samples, lake sediments, clean and oiled sand, and microbialites. Even though HAAL samples were obtained from mats and microbialites, other known microbialites grouped elsewhere.

## Discussion

This work reports the metagenomic analysis of two microbial mats from Atacama Desert, located in Brava and Tebenquiche lakes. These lakes have been studied previously [7, 12, 13, 63], describing physicochemical conditions and diverse microbial ecosystems. However, metabolic capabilities of the communities were only inferred from the taxonomic diversity obtained from 16S rRNA gene clone libraries or amplicon sequencing. Further insight was obtained

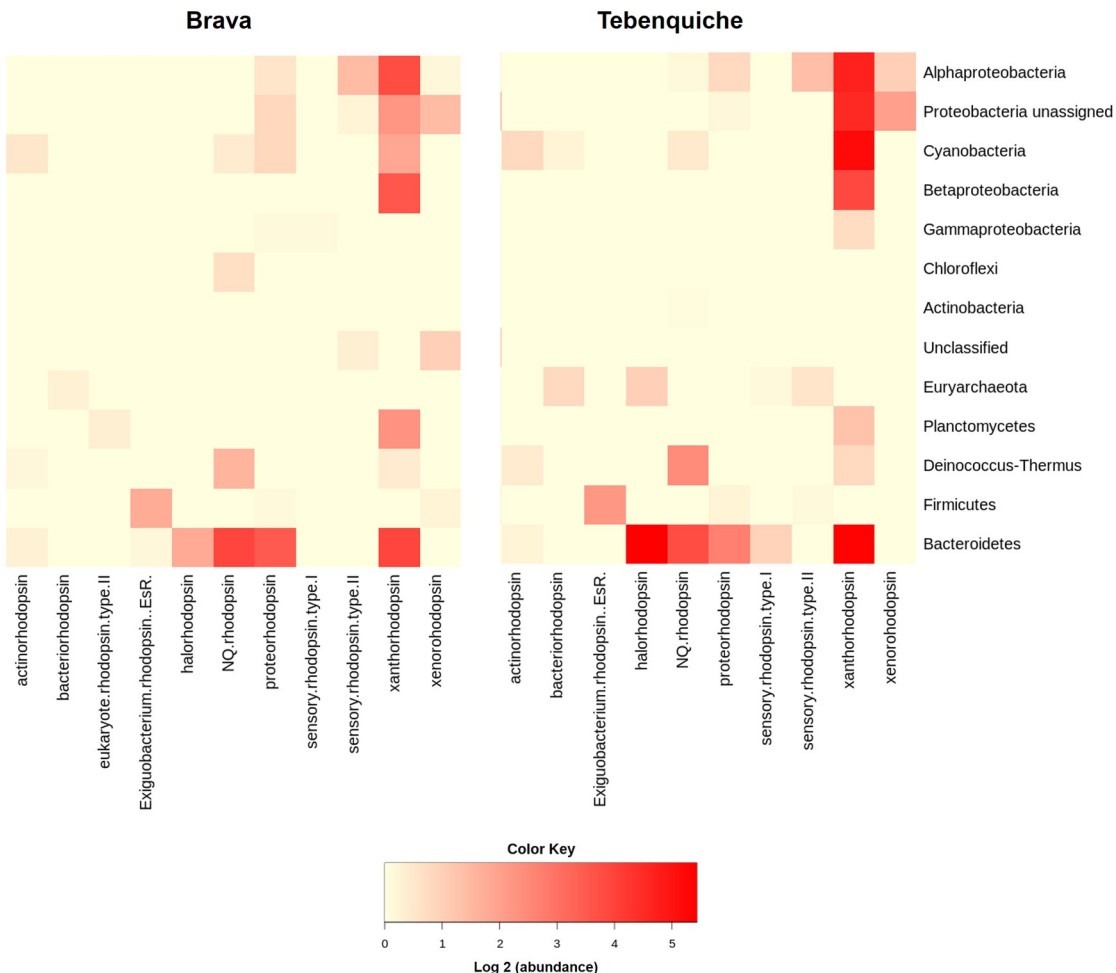

**Fig 5. Rhodopsin types, abundances and affiliation in Brava and Tebenquiche microbial mats.**

here through WGS sequencing of the two microbial mats, aiming to better describe them and understand the potential biogeochemical cycling in the mats.

The mats and their locations presented general patterns observed in previous studies [7, 12, 13, 63]. Briefly, water analyses showed hypersalinity, and a slightly basic pH (7.8). The taxonomic diversity was dominated by Proteobacteria and Bacteroidetes, particularly in Tebenquiche where they represented over 70% of total diversity. This taxonomic composition was in line with general observations in HAAL [4], and confirmed previous observations in mats from the same lakes [7].

An emerging feature of HAAL locations is the low abundance of Cyanobacteria [4], which was confirmed by our data. This raised the question about who are the primary producers in the ecosystem. Metagenomic data started to answer this by identifying genes involved in key processes and their affiliation. Genes for oxygenic photosynthesis were almost exclusively affiliated to Cyanobacteria in both mats. For Brava mat, Cyanobacteria abundance was over 10%, which was rather high compared to most HAAL locations, while in Tebenquiche mat it was around 5% (Fig 2). Even though these abundances could be enough to provide nutrients for the mats, signs of alternative primary producers could be found in the metagenomic data. Anoxygenic phototrops are always present in HAAL mats, and in these mats they were mostly

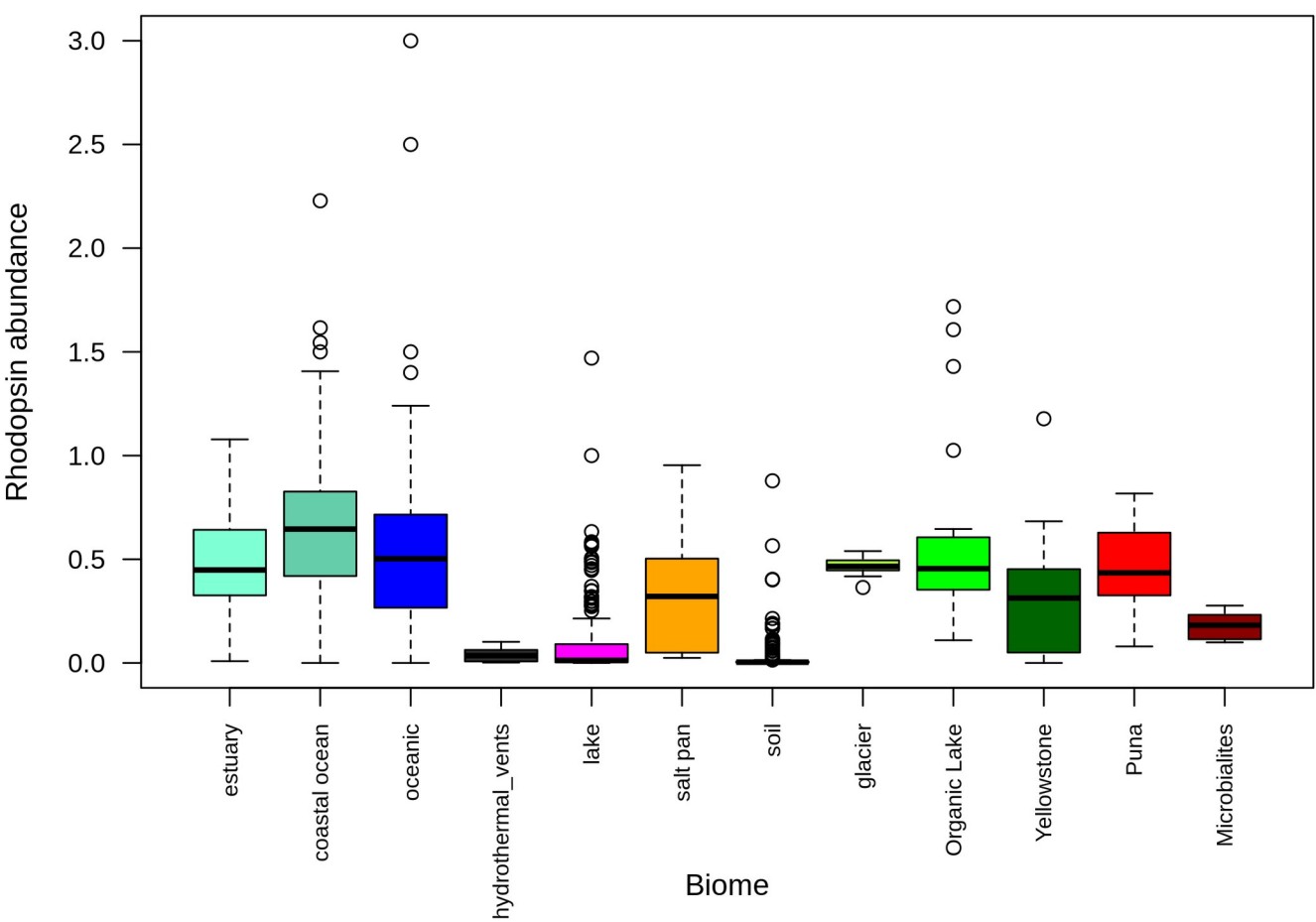

**Fig 6. Rhodopsin abundances in metagenomes.** Data from 1063 metagenomes, standardized to RecA and Rad51 abundances, and grouped by category Metagenomes from this work and other HAAL sites are included as "Puna".

represented by Alphaproteobacteria from the Rhizobiales and Rhodobacterales families, and also by Chloroflexi in Brava. Their importance could be seen comparing the relative abundances of the analyzed functions (S2 Table). In BM, with higher abundance of oxygenic photosynthesis genes, the other metabolisms are diminished, while in TM oxygenic photosynthesis is less abundant but all others are augmented. This could also be related to increased diversity and abundance of anaerobic organisms.

Elemental cycling is important in mats as a driver for lithification [74]. In both mats, sulfate reduction *apr* and *dsr* genes were mostly affiliated to Deltaproteobacteria and Firmicutes, and present in similar abundances, suggesting full reduction of sulfate up to $H_2S$. This pattern is also observed in Shark Bay mats [66]. On the other hand, sulfur oxidation affiliated mostly to Rhizobiales and Rhodobacterales, which could also be involved in anoxygenic photosynthesis using $H_2S$. This could be seen very roughly in TM (Fig 3), where the same majority phyla associated to both processes. Phyla affiliated to nitrogen fixation and denitrification were also similar, suggesting that related organisms might participate on both processes for each mat. Scarcity of ammonia oxidation genes has also been observed in Shark Bay mats [66] or Alchichica microbialites [75]. Alternative carbon fixation pathways were abundant, Calvin-Benson cycle was present but Wood-Ljundahl pathway had similar abundances. Moreover, from Calvin-Benson cycle representatives, only a fraction (34% Brava, 20% Tebenquiche) of the marker

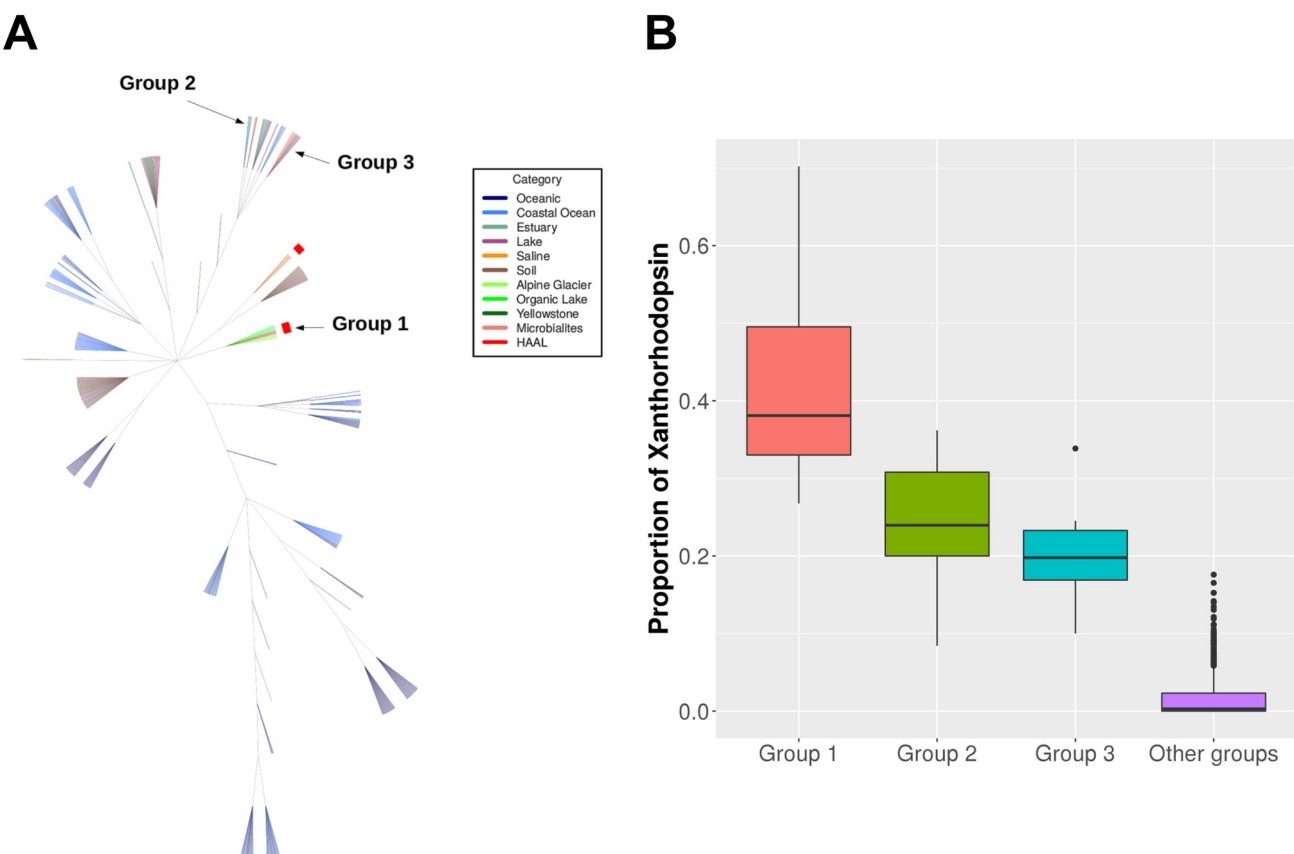

**Fig 7. Xanthorhodopsin clusters.** (A) Clustering of metagenomes from different environments based on rhodopsin type abundances. (B) Proportion of xanthorhodopsins in different groups.

genes was affiliated to *Cyanobacteria*. Thus, there is potential for carbon fixation to occur through alternative pathways/organisms. Actinobacteria could be one of those in Brava, even with low abundances, since 10% of Calvin-Benson markers are affiliated to them. In Antarctic soils, with low photosynthetic capacity, Actinobacteria and rare phyla fix carbon using atmospheric $H_2$ and CO [76]. In Atacama Desert soils, carbon fixing Actinobacteria have been detected [77]. Deltaproteobacteria represented about 50% of the organisms bearing the Wood-Ljundahl pathway (Fig 3B). Several Deltaproteobacteria can grow chemoautotrophically, relying on inorganic electron sources, such as molecular hydrogen [78]. It has been shown that in hypersaline microbial mats, they do consume $H_2$ [79], which would be produced by fermentative *Cyanobacteria*. Thus, given that these "alternative" producers depend on metabolites either directly ($H_2$, organic compounds) or indirectly ($H_2S$) originated by cyanobacterial activity, they might not be "primary" in a strict sense [80]. They could rely on atmospheric $H_2$ and CO, but their carbon turnover would be low. In modern day's hypersaline phototrophic mats, the contribution of oxygenic Cyanobacteria on the oxic zones is important and perhaps indispensable. This pattern of increasing abundance of alternative carbon fixation pathways and anoxygenic photosynthesis has been related to Early Earth analog environments [17], interestingly in Salar de Llamara, another location from Atacama Desert. The ancient Wood-Ljundahl pathway was observed in Brava and Tebenquiche mats, and it is also present in Socompa (another HAAL location), and in Shark Bay mats (Fig 4), which are also subjected to extreme conditions. Hypersalinity and high irradiation might be strongly influencing community

composition at both taxonomic and functional levels, generating similar communities in the HAAL and in Shark Bay (S2 Fig). Moreover, the high concentrations of arsenic in HAAL locations and evidence of active arsenic cycling in Brava [81] further support the idea that these mats could be used to model Early Earth conditions.

Even though it does not allow autotrophic growth, light transduction by microbial rhodopsins would be an alternative independent energy source [23, 82]. It has been shown that these proteins might be particularly important in extreme conditions [25, 83]. Rhodopsins were present in both mats, with most proteins being affiliated to Bacteroidetes. Given that these organisms are most likely heterotrophs, the ability to obtain additional energy from light could be a competitive advantage. In fact, this phylum is usually one of the most abundant in HAAL [4]. Rhodopsins have been found in other Andean metagenomes and isolates [18, 19, 73], including xantorhodopsin in *Salinivibrio* [72] and a functional proteorhodopsin in *Exiguobacterium* [71], both isolated from Socompa stromatolites [18, 19, 72, 73]. In Brava and Tebenquiche, several types of rhodopsins were found, but xanthorhodopsin was the most represented, followed by NQ rhodopsins in Brava and halorhodopsins in Tebenquiche (Fig 4). Actinorhodopsins were also found in our classification, affiliated to other phyla than Actinobacteria. However, these are likely unusual xanthorhodopsins that were wrongly typed due to the limited number of these proteins in Micrhode database. The abundance of these rhodopsin types points not only to energy generation, but also to ion extrusion as a survival mechanism [23, 62]. As an indirect functional role, it has been proposed that heterotrophic rhodopsin-bearing organisms are able to fix an increased fraction of the carbon as $CO_2$ through anaplerotic reactions in the presence of light [84], thus contributing to total carbon fixation.

Metagenome comparisons suggest that rhodopsins are important for microbial mats, where abundances are not as high as in oceanic metagenomes, but higher than in soil. Previous work had shown that microbial rhodopsins are mostly abundant in aquatic environments, although only a few non-aquatic environments were considered [28, 85]. Our results confirmed this trend, even with the inclusion of more soil environments and microbial mats (Fig 5). Comparison of the HAAL environments roughly points to altitude as an influencing factor. HAAL rhodopsin abundances are, in decreasing order: Socompa (3570 masl) > Tebenquiche (2274 masl) > Diamante (4589 masl) > Brava (2264 masl) > Llamara (754 masl). In Andean mats, a range of abundances were observed. This could be due to phototrophy being more important in some locations, or simply to variability in sampling, where communities from thicker mats included less phototrophic members, and thus lesser abundances. Disregarding total rhodopsin abundances, our results show that one of the hallmarks of Andean mats is the relative abundance of xantorhodopsins. This is not due to taxonomic similarities, as HAAL metagenomes do not cluster together when classified by phylum (S3 Fig), as can also be seen in their phyla relative abundances (S4 Fig). Xantorhodopsins are a diverse group of proteins and only a few have been biochemically characterized. The first proteins biochemically characterized displayed the ability to bind caroteinoid pigments, improving the wavelength range for absortion [26, 27]. Later studies pointed out two subgroups of the family, one of them unable to bind known cofactors as salinixanthin [28]. Still, they function as light-stimulated proton pumps, and it has been proposed that they would serve similar functions that marine proteorhodopsins, helping to survival in extreme conditions. In fact, HAAL environments are grouped with samples from the antartic hypersaline Organic lake [86], and surfaces from Baltoro (Pakistani Karakoram) and Forni (Italian Alps) glaciers [87], which could be deemed as extreme habitats. Moreover, xanthorhodopsins are abundant in other groups of metagenomes (Fig 7), including mats from Alchichica [75], Bahamas thrombolites [67], and sands [88]. Strikingly, xanthorhodopsins are not abundant in marine environments, and thus the reasons

for their selection in limited sites are intriguing. HAAL mat systems could be interesting models to provide insight on this subject.

## Conclusions

HAAL mat systems are unique extreme environments, as defined by some of their distinct physicochemical conditions and surprising biodiversity. However, only a few sites have been described by metagenomic analysis, and in terms of genes, studies have focused on arsenic metabolism. This work aimed to improve knowledge of these sites and focused on other characteristics that might be related to cope with the extreme conditions.

The study of two hypersaline microbial mats from Atacama Desert allowed to characterize their taxonomic and functional composition. Proteobacteria and Bacteroidetes were major phyla in these systems, and even though Cyanobacteria were assigned as the primary producers, their relative abundances were low. In spite of taxonomic differences, functional composition was similar on both systems.

Carbon fixation and rhodopsin mediated energy transduction were analyzed in order to investigate alternatives to primary production by Cyanobacteria, although metagenomic data was not enough to confirm this. However, unusual carbon fixation pathways were well represented, in contrast to other well studied mat systems, supporting the idea that the HAAL mats could be used, to some extent, to model Early Earth metabolisms.

Finally, rhodopsins from the xanthorhodopsin family are abundant not only on these mats, but seem to be a hallmark feature in high altitude Andean environments, as evidenced by comparison with other metagenomes. Besides the presence of an additional pigment for light harvesting, the advantage provided by this subfamily in an environmental setting is not clear, but its abundance in distinct settings could guide further work.

## Supporting information

**S1 Table. Assembly statistics for Brava and Tebenquiche mats metagenomes, and size distribution of their contigs.**
(XLSX)

**S2 Table. Calculated abundances for functional genes.** Data used in Fig 3.
(XLSX)

**S3 Table. Blast results against NCBI nr from proteins assigned to 4-hydroxybutyryl-CoA dehydratase (K14534).**
(XLSX)

**S4 Table. Access identifiers and assigned biomes for the 1063 metagenomes analyzed in this work.**
(XLSX)

**S1 Fig. Classification of protein fragments.** X being the fragment to be classified, it will belong to a certain protein family if its score and coverage are greater than the **c** quantile of the score (S) and coverage (C) of the set of positive fragments, which have the length of X and the extremes of their alignment contain those of X, or are contained by those of X.
(TIF)

**S2 Fig. Principal component analysis at phylum, genus and metabolism (COG) levels for selected metagenomes.**
(TIF)

**S3 Fig. Clustering of metagenomes from different environments based on phylum abundances.**
(TIF)

**S4 Fig. Phylum abundances for high altitude Andean metagenomes.**
(TIF)

**S1 File.**
(DOCX)

## Acknowledgments

The authors thank Marco Contreras for his support during the sampling campaign and Geol. Jorge L. Rasuk MAusIMM for his assistance with the map figure. Map data available from the U.S. Geological Survey.

## Author Contributions

**Conceptualization:** Daniel Kurth, Manuel Contreras, María Eugenia Farías.

**Data curation:** Daniel Kurth, Dario Elias.

**Formal analysis:** Daniel Kurth, Dario Elias.

**Funding acquisition:** Manuel Contreras, María Eugenia Farías.

**Investigation:** Daniel Kurth, Dario Elias, María Cecilia Rasuk, Manuel Contreras, María Eugenia Farías.

**Methodology:** Daniel Kurth, Dario Elias.

**Resources:** María Cecilia Rasuk, Manuel Contreras, María Eugenia Farías.

**Software:** Daniel Kurth, Dario Elias.

**Supervision:** María Eugenia Farías.

**Validation:** Dario Elias.

**Visualization:** Daniel Kurth, Dario Elias, María Cecilia Rasuk.

**Writing – original draft:** Daniel Kurth, Dario Elias, María Eugenia Farías.

**Writing – review & editing:** Daniel Kurth, María Cecilia Rasuk, María Eugenia Farías.

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
