## [Decision Letter · Decision Letter 0]

20 Apr 2020

PONE-D-20-05616

Carbon fixation and rhodopsin systems in microbial mats from hypersaline lakes Brava and Tebenquiche, Salar de Atacama, Chile

PLOS ONE

Dear Dr. Farias,

Thank you for submitting your manuscript to PLOS ONE. I have received only one reviewer report, so far. However, in order to speed up the process, I send you my decision of major revision requested for the  publication of the manuscript. In the second round of revision, I will try to assure a second reviewer report.  Therefore, we invite you to submit a revised version of the manuscript that addresses the points raised during the review process, particularly include a more robust analysis of rhodopsin abundance.

We would appreciate receiving your revised manuscript by Jun 04 2020 11:59PM. To enhance the reproducibility of your results, we recommend that if applicable you deposit your laboratory protocols in protocols.io, where a protocol can be assigned its own identifier (DOI) such that it can be cited independently in the future. For instructions see: http://journals.plos.org/plosone/s/submission-guidelines#loc-laboratory-protocols

We look forward to receiving your revised manuscript.

Kind regards,

Andrea Franzetti

Academic Editor

PLOS ONE

Journal Requirements:

Reviewers' comments:

Reviewer's Responses to Questions

**Comments to the Author**

1. Is the manuscript technically sound, and do the data support the conclusions?

Reviewer #1: Yes

2. Has the statistical analysis been performed appropriately and rigorously? 

Reviewer #1: N/A

3. Have the authors made all data underlying the findings in their manuscript fully available?

Reviewer #1: Yes

4. Is the manuscript presented in an intelligible fashion and written in standard English?

Reviewer #1: Yes

5. Review Comments to the Author

Reviewer #1: On reviewing this manuscript, I conclude that this study begins to shed light into a very intriguing question regarding who the primary producers are in ecosystems such as microbial mats that contain low abundances of Cyanobacteria. The study, using two non-lithifying microbial mats collected from the lakes in the Salar de Atacama, Chile utilizes WGS sequencing to build a ‘case’ for microbial rhodopsins as an alternative, independent energy source. The study compared multiple mats, from the stromatolites in Shark Bay, Australia to the high- altitude mats from Socompa Lake, in the Andean plateau, as well as the stromatolites from the open marine enviromment of Highbourne Cay, The Bahamas in an effort to answer this question.

Although robust in taxonomic classification and reporting the COGs of carbon fixation, the study lacked a thorough analysis of diverisity metrics and rhodopsin abundance of these other environments included in the study. I will provide recommendations for a more robust analysis before recommending a full acceptance.

Overall recommendations:

1) All Figures – recommend to increase the resolution of all figures as they were extremely blurry and hard to see when zooming in. At 100% zoom, quite a few figures (e.g. figure 5), the text was very hard to read.

2) Addition of a conclusion paragraph - there was no final conclusion statement – the manuscript abruptly ends (line 422), prompting me to search if I was missing a page. Recommend to add a conclusion paragraph that states the research question, results found supporting the study, and overall conclusions from the study.

3) Addition of minimal arsenic data - this manuscript mentions that these mats grow under high levels of ‘toxic metals’ and metaloids and the authors continuously mention arsenic in multiple places (e.g. introduction, line 52, 378-379; however, the study is primarily focused on carbon fixation and rhodopsin. Due to the fact that this is mentioned often, suggest putting in some minimal arsenic data (annotated arsenic genes, etc) to show that arsenic may indeed be tied to altitude – such as the same authors suggested in their 2017 study of arsenic metabolism in the HAAL system of Socompa.

4) Beta Diversity metrics – recommend to add an analysis of all sites together; explanation below.

5) Rhodopsin abundance per site – recommend to include an RHO abundance/site to corroborate if RHO is truly in higher abundances in HAAL environments. This result has the potential to positively influence the hypothesis of the study.

Other specific comments to follow.

Abstract

Line 22: Please define WGS since it’s appearing for the first time in this manuscript.

Introduction

The introduction was very clear. It included a lot of background information and the purpose of the study was clear – here are some recommendations for all sections of the manuscript:

Line 46: What is the altitude of the study sites? You mention this in the discussion, but adding altitude here would also be beneficial, especially since altitude is being mentioned.

Lines 52-56: 1) Characterizes HAAL and is comparing this environment as similar to where the samples were collected. Suggest adding in at least another environment that is ‘extreme’ for comparison purposes, to avoid such a dichotomous comparison, such as the environment of Shark Bay, which is similar due to hyper salinity, etc. and it is also used for comparison purposes in this study.

2) Also, what is the altitude for the Puna Andrean region (line 53)?

Line 58: characterize taxonomically to: taxonomically characterize

Materials and Methods

Line 105: Samples were pooled prior to sequencing – how many rounds of DNA extraction were conducted with the Power Biofilm Kit? Was 1 enough? (microbial mats are known to be rich in EPS).

Line 116: What was the percent retained after paired end reads were merged?

Line 118: There is no coverage information given – suggest to include some coverage statistics (e.g. coverage/contig, average coverage, etc).

Lines 122-125: 1) What alpha diversity metrics were used to assess the data? Line 235 states that alpha diversity had an observed value of 572 – what metric was used?

2) How was processing and filtering of data conducted (e.g. low abundance)?

Line 152 – 153: States – “Brava and Tebenquiche, and preprocess was required including adaptor trimming and quality filtering with Trimmomatic” Recommend changing to: …. and preprocessing, which included adaptor trimming and quality filtering ….

Line 169: In Supplemental Fig. 1: Suggest changing ‘X belong’ to ‘X belongs’.

Line 193: Please cite R. Run the command as is: citation() in R to show how to cite R.

Line 205-211: Calculating RHO abundance – was this calculation generated in house? Or was this borrowed from another publication? Your cite of [53] didn’t seem to include this RHO abundance ratio. If this ratio is noted elsewhere in the literature, please cite where it came from.

Results

For the ‘Taxonomic Diversity’ section, this section would be enhanced greatly if all metagenomes collected (Brava, Tebenquiche, Cuatro Cienegas, Diamante, Shark Bay Smooth and Pustular mats, HBC thrombolites, Socompa, Cape Recite, Schoenmakerskop, S. Africa, Markham Ice Shelf, Ward Hunt Ice Shelf, and McMurdo) were visually compared under a beta diversity metric (e.g. PCA, NMDS, PCoA, etc. with appropriate statistical testing) so that the following can be extrapolated: 1) percent variance and; 2) a visual representation of the ‘groupings’. This may be very informative to see the clustering patterns – if high altitude clusters together with Shark Bay mats or if there is a gradient of clustering on a metric. This might give clues as to the statements made in lines 283-284 about how the Wood-Ljungdahl were only observed in HAAL mats and in Shark Bay, at least in a taxonomic context.

Line 223-224: Mentions that Kaiju assigned 45% and 51% of the reads. Suggest that MG-RAST assignment is also placed.

Line 243: Please remove the word ‘transcript’ in TPM. Unless this was an RNA-Seq analysis. Suggest replacing with reads per million or counts per million.

Line 251: Please define BM and TM, since it’s the first time appearing.

Line 269: What is considered ‘poor number of hits’?

Line 277: suggest changing to: finally, 4-hydroxybutyryl-CoA dehydratase (K14534), the key enzyme

Line 284: A note: Shark Bay mats are known to have this pathway via previous WGS in publications.

Line 284: For Figure 4: Suggest delineating the classifications of each mat in the arbitrary grouping that is mentioned in line 148-149 or via environmental parameters (e.g. hypersaline, high altitude, etc.) to guide the reader as to what environment these mats are from.

Line 294: Suggest changing to: 182 and 253 coding sequences were found for rhodopsins in the metagenomic assemblies.

Line 300: ‘proteins; are mentioned – suggest indicating what proteins this refers to.

Line 324: Figure 6 is unclear and confusing. With increasing metagenomes, it is assumed Rhodopsin abundance will also increase.

This section would be greatly enhanced if data is also reported similarly as in Figure 4. For each of the locations in Figure 4, report what the RHO abundances are (as the Y axis). This may help to corroborate the thesis of this paper - if RHO is truly in higher abundances in HAAL environments (and perhaps even Shark Bay), where cyanobacterial abundances are low or lower than other microbial mat ecosystems. This result has the potential to have a strong influence on this area of study.

End of Edit.

Good Luck.

6. PLOS authors have the option to publish the peer review history of their article (what does this mean?). If published, this will include your full peer review and any attached files.

Reviewer #1: No

---

## [Author Response · Author response to Decision Letter 0]

27 Oct 2020

We appreciate the reviewer suggestions as they improved the manuscript, allowing us to correct several flaws.

Overall recommendations:

1) All Figures – recommend to increase the resolution of all figures as they were extremely blurry and hard to see when zooming in. At 100% zoom, quite a few figures (e.g. figure 5), the text was very hard to read.

Some figures were changed and resolution improved.

2) Addition of a conclusion paragraph - there was no final conclusion statement – the manuscript abruptly ends (line 422), prompting me to search if I was missing a page. Recommend to add a conclusion paragraph that states the research question, results found supporting the study, and overall conclusions from the study.

A conclusions section has been added

3) Addition of minimal arsenic data - this manuscript mentions that these mats grow under high levels of ‘toxic metals’ and metaloids and the authors continuously mention arsenic in multiple places (e.g. introduction, line 52, 378-379; however, the study is primarily focused on carbon fixation and rhodopsin. Due to the fact that this is mentioned often, suggest putting in some minimal arsenic data (annotated arsenic genes, etc) to show that arsenic may indeed be tied to altitude – such as the same authors suggested in their 2017 study of arsenic metabolism in the HAAL system of Socompa.

Arsenic is a major subject for the HAAL locations, and a comparative study including the data from this work is being carried out. A preprint is available and has been cited in the text (Line 86)

4) Beta Diversity metrics – recommend to add an analysis of all sites together; explanation below.

The analysis suggested by the reviewer was performed and added to the manuscript.

5) Rhodopsin abundance per site – recommend to include an RHO abundance/site to corroborate if RHO is truly in higher abundances in HAAL environments. This result has the potential to positively influence the hypothesis of the study.

Rhodopsin abundance was actually calculated and shown in figure 6. The result was poorly presented in the original manuscript and the figure has been changed in the revised version.

Other specific comments to follow.

Abstract

Line 22: Please define WGS since it’s appearing for the first time in this manuscript.

Fixed.

Introduction

The introduction was very clear. It included a lot of background information and the purpose of the study was clear

– here are some recommendations for all sections of the manuscript:

Line 46: What is the altitude of the study sites? You mention this in the discussion, but adding altitude here would also be beneficial, especially since altitude is being mentioned.

Data added.

Lines 52-56: 1) Characterizes HAAL and is comparing this environment as similar to where the samples were collected. Suggest adding in at least another environment that is ‘extreme’ for comparison purposes, to avoid such a dichotomous comparison, such as the environment of Shark Bay, which is similar due to hyper salinity, etc and it is also used for comparison purposes in this study.

Actually, the intention was to include the study sites within the broad HAAL category.

2) Also, what is the altitude for the Puna Andrean region (line 53)?

Data added.

Line 58: characterize taxonomically to: taxonomically characterize

Fixed.

Materials and Methods

Line 105: Samples were pooled prior to sequencing – how many rounds of DNA extraction were conducted with the Power Biofilm Kit? Was 1 enough? (microbial mats are known to be rich in EPS).

Added to text: “One round of extraction and pooling allowed obtaining enough DNA (pooled samples included DNA from three extractions, one from each triplicate core).”

Line 116: What was the percent retained after paired end reads were merged?

This was performed by the sequence provider and we do not have that data. Still, around 90% of the reads could be mapped to the assembly.

Line 118: There is no coverage information given – suggest to include some coverage statistics (e.g. coverage/contig, average coverage, etc).

Some data was added to S1 Table.

Lines 122-125: 1) What alpha diversity metrics were used to assess the data? Line 235 states that alpha diversity had an observed value of 572 – what metric was used?

This metric was provided by MG-RAST server, and it’s based on protein taxonomic annotations at the species level. A short paragraph was added to Materials and methods (see lines 131-135), describing the metric used.

2) How was processing and filtering of data conducted (e.g. low abundance)?

There was no processing other than performed by the MG-RAST server. Most analysis involve identifying specific features. Taxonomic analysis was rather superficial, only to present a broad view of the diversity, and in the figure 2 low abundance data (<1%) was grouped as "Minor phyla"

Line 152 – 153: States – “Brava and Tebenquiche, and preprocess was required including adaptor trimming and quality filtering with Trimmomatic” Recommend changing to: …. and preprocessing, which included adaptor trimming and quality filtering ….

Fixed.

Line 169: In Supplemental Fig. 1: Suggest changing ‘X belong’ to ‘X belongs’.

Fixed.

Line 193: Please cite R. Run the command as is: citation() in R to show how to cite R.

Fixed.

Line 205-211: Calculating RHO abundance – was this calculation generated in house? Or was this borrowed from another publication? Your cite of [53] didn’t seem to include this RHO abundance ratio. If this ratio is noted elsewhere in the literature, please cite where it came from.

Two citations added: 

Sharma AK, Zhaxybayeva O, Papke RT, Doolittle WF. Actinorhodopsins: Proteorhodopsin-like gene sequences found predominantly in non-marine environments. Environ Microbiol. 2008;10: 1039–1056. doi:10.1111/j.1462-2920.2007.01525.x

Kwon S-K, Kim BK, Song JY, Kwak M-J, Lee CH, Yoon J-H, et al. Genomic makeup of the marine flavobacterium Nonlabens (Donghaeana) dokdonensis and identification of a novel class of rhodopsins. Genome Biol Evol. 2013;5: 187–99. doi:10.1093/gbe/evs134

Results

For the ‘Taxonomic Diversity’ section, this section would be enhanced greatly if all metagenomes collected (Brava,Tebenquiche, Cuatro Cienegas, Diamante, Shark Bay Smooth and Pustular mats, HBC thrombolites, Socompa, Cape Recite, Schoenmakerskop, S. Africa, Markham Ice Shelf, Ward Hunt Ice Shelf, and McMurdo) were visually compared under a beta diversity metric (e.g. PCA, NMDS, PCoA, etc. with appropriate statistical testing) so that the following can be extrapolated: 1) percent variance and; 2) a visual representation of the ‘groupings’. This may be very informative to see the clustering patterns – if high altitude clusters together with Shark Bay mats or if there is a gradient of clustering on a metric. This might give clues as to the statements made in lines 283-284 about how the Wood-Ljungdahl were only observed in HAAL mats and in Shark Bay, at least in a taxonomic context.

The suggested analysis was performed, see Metagenomic comparison sections at Materials and methods, and the Taxonomic and functional diversity section at Results.

Line 223-224: Mentions that Kaiju assigned 45% and 51% of the reads. Suggest that MG-RAST assignment is also placed.

Fixed.

Line 243: Please remove the word ‘transcript’ in TPM. Unless this was an RNA-Seq analysis. Suggest replacing with reads per million or counts per million.

Fixed.

Line 251: Please define BM and TM, since it’s the first time appearing.

Fixed.

Line 269: What is considered ‘poor number of hits’?

This meant a normalized abundance below 1 cpm, which would be barely visible on Fig 3B. The explanation was added to the text.

Line 277: suggest changing to: finally, 4-hydroxybutyryl-CoA dehydratase (K14534), the key enzyme

Fixed.

Line 284: A note: Shark Bay mats are known to have this pathway via previous WGS in publications.

Yes, the following reference was added:

Wong HL, White RA, Visscher PT, Charlesworth JC, Vázquez-Campos X, Burns BP. Disentangling the drivers of functional complexity at the metagenomic level in Shark Bay microbial mat microbiomes. ISME J. 2018;12: 2619–2639. doi:10.1038/s41396-018-0208-8

Line 284: For Figure 4: Suggest delineating the classifications of each mat in the arbitrary grouping that is mentioned in line 148-149 or via environmental parameters (e.g. hypersaline, high altitude, etc.) to guide the reader as to what environment these mats are from.

This was briefly mentioned in the text and added to Figure 4.

Line 294: Suggest changing to: 182 and 253 coding sequences were found for rhodopsins in the metagenomic assemblies.

Fixed.

Line 300: ‘proteins; are mentioned – suggest indicating what proteins this refers to.

Changed “proteins” to “sequences”.

Line 324: Figure 6 is unclear and confusing. With increasing metagenomes, it is assumed Rhodopsin abundance will also increase.

The figure presented all analyzed metagenomes sorted by abundance. “Increasing metagenomes” did not mean the sum of the abundances. We acknowledge that it was confusing and the figure was changed.

This section would be greatly enhanced if data is also reported similarly as in Figure 4. For each of the locations in Figure 4, report what the RHO abundances are (as the Y axis). This may help to corroborate the thesis of this paper - if RHO is truly in higher abundances in HAAL environments (and perhaps even Shark Bay), where cyanobacterial abundances are low or lower than other microbial mat ecosystems. This result has the potential to have a strong influence on this area of study

The abundance analysis involved hundreds of metagenomes, and thus is poorly presented in a bar graph as in Figure 4. The original Figure 6 was actually a bar graph. Thus, metagenomes were arbitrarily grouped (as already described in text) and a boxplot is presented, summarizing abundances observed for the different groups.

---

## [Decision Letter · Decision Letter 1]

11 Dec 2020

PONE-D-20-05616R1

Carbon fixation and rhodopsin systems in microbial mats from hypersaline lakes Brava and Tebenquiche, Salar de Atacama, Chile

PLOS ONE

Dear Dr. Farias,

Thank you for submitting your manuscript to PLOS ONE. After careful consideration, we feel that it has merit but does not fully meet PLOS ONE’s publication criteria as it currently stands. Therefore, we invite you to submit a revised version of the manuscript that addresses the points raised during the review process.

Please address all the points raised by the referee, with particular focus to the request of including the discussion of other metabolisms.

We look forward to receiving your revised manuscript.

Kind regards,

Andrea Franzetti

Academic Editor

PLOS ONE

Reviewers' comments:

Reviewer's Responses to Questions

**Comments to the Author**

1. If the authors have adequately addressed your comments raised in a previous round of review and you feel that this manuscript is now acceptable for publication, you may indicate that here to bypass the “Comments to the Author” section, enter your conflict of interest statement in the “Confidential to Editor” section, and submit your "Accept" recommendation.

Reviewer #2: (No Response)

2. Is the manuscript technically sound, and do the data support the conclusions?

Reviewer #2: Yes

3. Has the statistical analysis been performed appropriately and rigorously? 

Reviewer #2: Yes

4. Have the authors made all data underlying the findings in their manuscript fully available?

Reviewer #2: Yes

5. Is the manuscript presented in an intelligible fashion and written in standard English?

Reviewer #2: Yes

6. Review Comments to the Author

Reviewer #2: The study is of interest, metagenomic information on these kinds of hypersaline environments and microbial mats is still scarce in the public record. In data analysis, the authors stepped further from the MG-RAST analysis, and additionally evaluated genes of interest and compared their distribution with other environments worldwide. These observations are the real strength of the study. The authors also made their data publicly available in different web servers.

Major remarks:

However I understand that it was not the main goal of the study, but I would have liked to read more about other metabolisms detected in the mat metagenomes as well (e.g. N-utilization, only mentioned in Ln 285) in the results and discussion sections to understand more the local nutrient cycle.

How could the authors explain that the major proportions of actinorhodopsines detected (Fig 5) were not affiliated with Actinobacteria at all (contrary to Sharma et al., 2008)?

I had a few additional remarks, but these are rather suggestions for future investigations than criticism on the present work:

Metagenomes from more than one site per lake could have strengthened the assumptions; it would be interesting to also compare metagenomes of the communities of the water and the sediment with mat microbial communities. Based on the data provided in Suppl Table 1, MAG assembly could have been carried out using easy to handle softwares such as SqueezeMeta (Tamames ans Sanchez 2019 Front Microbiol 9:3349). The comparison of individual MAGs rather than the community metabolic potential could have revealed more detailed differences in microbial metabolisms. Low abundance of Cyanobacteria in the metagenomes is an interesting observation, but it could raise a few doubts (could be due to biased cell wall disruption, high amount of EPS during DNA isolation?). It would be beneficial to validate the results with microscopic observations (autofluorescent cells/total number of cells).

Minor remarks:

- the authors confusingly put phyla/class names in italic in some places in the text (e.g. Ln 248-288, 337-338, 394) and in other paragraphs are not.

Please correct the following typos:

Ln 237: correct gigabases (Gb) to megabases (Mbp) as in Suppl Table 1, bp is missing after 1394.

Additionally, I think base pair (bp) is more sound in English than pair base (pb).

Ln 293 "Other well-known anoxygenic photosynthesizers also fix carbon through Calvin-Benson", please extend the latter with "cycle/pathway"

Ln 311 space is missing in "Thepresence of Wood-Ljungdahl pathway"

Ln 407 correct to "Shark Bay"

Ln 434 correct to "phototrophic"

---

## [Author Response · Author response to Decision Letter 1]

22 Jan 2021

Quoted reviewer comments and our answers below. We appreciate the reviewer suggestions as they improved the manuscript, allowing us to correct several flaws.

"The study is of interest, metagenomic information on these kinds of hypersaline environments and microbial mats is still scarce in the public record. In data analysis, the authors stepped further from the MG-RAST analysis, and additionally evaluated genes of interest and compared their distribution with other environments worldwide. These observations are the real strength of the study. The authors also made their data publicly available in different web servers.

Major remarks:

However I understand that it was not the main goal of the study, but I would have liked to read more about other metabolisms detected in the mat metagenomes as well (e.g. N-utilization, only mentioned in Ln 285) in the results and discussion sections to understand more the local nutrient cycle."

The results section on Taxonomic and Functional diversity was modified to better present these metabolisms (lines 265-301), and a Supplementary table (S2 table) was added. The discussion section was also improved (lines 417-430). However, as the reviewer states in other comments, additional analysis such as MAGs would be required to obtain further insight.

"How could the authors explain that the major proportions of actinorhodopsines detected (Fig 5) were not affiliated with Actinobacteria at all (contrary to Sharma et al., 2008)?"

We had not considered this issue. This could be a limitation of the MicRhode database. More details on rhodopsin classification were added to methods (lines 156-160). Rhodopsin classification was based on blast best hits to MicRhode. We used the last version available (05-12-2014). However, most of its proteins are classified as proteorhodopsins, and there are only 13 xanthorhodopsins and 267 actinorhodopsins. Given that actinorhodopsins are closely related to xanthorhodopsins and could be considered a subtype of this group, we propose that some of these “actinorhodopsins” from Fig 5 could actually be xanthorhodopsins. In fact, considering 5 blast hits, xanthorhodopsins show up among all of them. Xanthorhodopsin diversity is underrepresented in the database, and HAAL sites could include “borderline” xantho/actinorhodopsins.

"I had a few additional remarks, but these are rather suggestions for future investigations than criticism on the present work:

Metagenomes from more than one site per lake could have strengthened the assumptions; it would be interesting to also compare metagenomes of the communities of the water and the sediment with mat microbial communities. Based on the data provided in Suppl Table 1, MAG assembly could have been carried out using easy to handle softwares such as SqueezeMeta (Tamames ans Sanchez 2019 Front Microbiol 9:3349). The comparison of individual MAGs rather than the community metabolic potential could have revealed more detailed differences in microbial metabolisms. Low abundance of Cyanobacteria in the metagenomes is an interesting observation, but it could raise a few doubts (could be due to biased cell wall disruption, high amount of EPS during DNA isolation?). It would be beneficial to validate the results with microscopic observations (autofluorescent cells/total number of cells)."

Your suggestions are welcomed and appreciated. We are more interested in the whole HAAL region, looking for similarities in its varied niches. We agree that a more in depth study of these sites could have included water and sediment communities, MAG assemblies, and obviously more samples. Low abundance of Cyanobacteria is a remarkable HAAL feature, and we had attempted your suggestions in earlier studies from different sites, such as Socompa lake. There, even though Cyanobacteria could be easily observed at the microscope, the diversity and abundance of other phyla is overwhelming. Still, that analysis was not performed for the current sites and is a shortfall of this work.

"Minor remarks:

- the authors confusingly put phyla/class names in italic in some places in the text (e.g. Ln 248-288, 337-338, 394) and in other paragraphs are not."

This was corrected.

"Please correct the following typos:

Ln 237: correct gigabases (Gb) to megabases (Mbp) as in Suppl Table 1, bp is missing after 1394. Additionally, I think base pair (bp) is more sound in English than pair base (pb).

Ln 293 "Other well-known anoxygenic photosynthesizers also fix carbon through Calvin-Benson", please extend the latter with "cycle/pathway"

Ln 311 space is missing in "Thepresence of Wood-Ljungdahl pathway"

Ln 407 correct to "Shark Bay"

Ln 434 correct to "phototrophic""

All these typos were corrected.

---

## [Editor Report · Decision Letter 2]

25 Jan 2021

Carbon fixation and rhodopsin systems in microbial mats from hypersaline lakes Brava and Tebenquiche, Salar de Atacama, Chile

PONE-D-20-05616R2

Dear Dr. Farias,

We’re pleased to inform you that your manuscript has been judged scientifically suitable for publication and will be formally accepted for publication once it meets all outstanding technical requirements.

Kind regards,

Andrea Franzetti

Academic Editor

PLOS ONE
---

## [Editor Report · Acceptance letter]

29 Jan 2021

PONE-D-20-05616R2 

Carbon fixation and rhodopsin systems in microbial mats from hypersaline lakes Brava and Tebenquiche, Salar de Atacama, Chile 

Dear Dr. Farías:

I'm pleased to inform you that your manuscript has been deemed suitable for publication in PLOS ONE. Congratulations! Your manuscript is now with our production department. 

Kind regards, 

on behalf of

Dr. Andrea Franzetti 

Academic Editor

PLOS ONE